# The Potential of Mesenchymal Stem Cells in Treating Spinocerebellar Ataxia: Advances and Future Directions

**DOI:** 10.3390/biomedicines12112507

**Published:** 2024-11-01

**Authors:** Gi Beom Lee, Se Min Park, Un Ju Jung, Sang Ryong Kim

**Affiliations:** 1School of Life Science and Biotechnology, BK21 FOUR KNU Creative BioResearch Group, Kyungpook National University, Daegu 41566, Republic of Korea; gibuom7@naver.com (G.B.L.); qkrtpals89@naver.com (S.M.P.); 2Department of Food Science and Nutrition, Pukyong National University, Busan 48513, Republic of Korea; jungunju@pknu.ac.kr; 3Brain Science and Engineering Institute, Kyungpook National University, Daegu 41404, Republic of Korea

**Keywords:** spinocerebellar ataxia, mesenchymal stem cells, immunomodulation, homing ability

## Abstract

Spinocerebellar ataxia (SCA) is a heterogeneous disorder characterized by impaired balance and coordination caused by cerebellar dysfunction. The absence of treatments approved by the U.S. Food and Drug Administration for SCA has driven the investigation of alternative therapeutic strategies, including stem cell therapy. Mesenchymal stem cells (MSCs), known for their multipotent capabilities, have demonstrated significant potential in treating SCA. This review examines how MSCs may promote neuronal growth, enhance synaptic connectivity, and modulate brain inflammation. Recent findings from preclinical and clinical studies are also reviewed, emphasizing the promise of MSC therapy in addressing the unmet needs of SCA patients. Furthermore, ongoing clinical trials and future directions are proposed to address the limitations of the current approaches.

## 1. Introduction

Cerebellar ataxia (CA) is a heterogeneous disorder characterized by difficulties in balance and coordination resulting from cerebellar dysfunction [1,2]. This dysfunction can lead to various symptoms, including dysarthria and nystagmus. Currently, no definitive treatments are available for CA, with management primarily focusing on symptom relief rather than addressing root causes. Among the different forms of CA, spinocerebellar ataxia (SCA) is classified as an autosomal dominant disorder, with treatment further complicated by multiple mutation types [3]. SCA comprises several subtypes primarily classified by genetic mutations, with polyglutamine repeat expansion ataxias being the most prevalent. These mutations lead to progressive neurodegeneration, particularly in the cerebellum [4]. Consequently, researchers have been motivated to explore innovative therapeutic approaches, such as stem-cell-based therapies, to provide more effective treatment options for SCA patients. Mesenchymal stem cells (MSCs) are actively being investigated as a potential therapy for SCA and various other neurological diseases.

MSCs possess unique characteristics that distinguish them from other types of stem cells. These multipotent stem cells can differentiate into various functional cell types, self-renew, and be isolated from different sources, such as adipose tissue, umbilical cords, and bone marrow. Additionally, MSCs can suppress immune responses and promote recovery by modulating adjacent immune cells through paracrine activity. The therapeutic application of MSCs in SCA animal models is being extensively assessed in preclinical studies [5,6,7,8,9]. Experimental results have determined that MSCs can successfully migrate to the cerebellum in SCA mouse models, secreting neurotrophic factors that promote the survival of Purkinje cells (PCs) [5]. Interest in the clinical utility of MSCs is growing rapidly. Given that CA diagnosis and treatment can vary depending on the underlying causes, this review provides a summary of SCA and discusses their common causes. Furthermore, stem-cell-based therapy is reviewed as a novel treatment strategy for SCA. First, we provide an overview of stem cell therapy, focusing on MSCs. Next, we explore the use of MSCs in other neurological diseases and their application in SCA mouse models. Finally, both ongoing and completed clinical trials involving MSCs for SCA therapy are reviewed. This comprehensive review aims to emphasize the potential of MSC therapy to address unmet needs in SCA treatment and to offer insights into future research directions.

## 2. Overview of SCAs

Ataxia can be inherited, acquired, or sporadic (non-hereditary degenerative ataxias), with SCA being a major subtype exhibiting autosomal dominant inheritance [10,11,12]. The pathological mechanism of SCAs involves the degeneration of PCs in the cerebellum and neuronal loss in other brain regions, including the brainstem, spinal cord, and deep cerebellar nuclei [13,14]. This degeneration disrupts essential motor coordination pathways and affects cerebellar regions connected to the limbic system, leading to a range of both motor and non-motor symptoms. These symptoms include gait instability, dysmetria, dysarthria, nystagmus, tremors, respiratory, cognitive dysfunction, and depression [13,14,15]. In addition to autosomal dominant ataxia, hereditary ataxias can be further classified into autosomal recessive, X-linked, mitochondrial, and episodic ataxias [1]. Among the autosomal recessive ataxias, Friedreich’s ataxia is the most common, caused by a homozygous intronic GAA triplet expansion in the FXN gene [16,17].

Fifty SCA types can be categorized based on mutated genes in polyglutamine repeat expansions, non-coding repeat expansions, and point mutations [3]. Polyglutamine (Poly Q) SCAs, the most common subtype, are caused by CAG-repeat expansions that encode polyglutamine tracts, including SCA1, SCA2, SCA3, SCA6, SCA7, and SCA17. In SCA17, the repeat organization of [(CAG)^3^ (CAA)^2^] and [CAA (CAG)_n_ CAA CAG] has been proposed [2,3,18]. Poly Q SCAs are characterized by an earlier onset of disease as the length of the CAG repeat increases [19,20]. The instability of CAG repeats and the open state of DNA contribute to longer repeat lengths across generations, resulting in earlier disease onset and more severe symptoms—a phenomenon known as “anticipation.” In contrast, non-repeat expansion SCAs generally manifest at a younger age and present a combination of cerebellar and non-cerebellar symptoms, including paroxysmal episodes such as seizures, hemiplegic migraines, and dystonia [21,22].

The pathological mechanisms underlying ataxia are highly complex, with the most common cause being the degeneration of PCs in the cerebellum. PCs play a crucial role in modulating the cerebellar output to other brain regions, and their degeneration disrupts motor coordination and balance [23,24,25]. In addition to PC loss, degeneration of the deep cerebellar nuclei, which serve as relay stations for cerebellar output, further exacerbates motor dysfunction.

Other affected regions include the cerebellar peduncles, essential for transmitting information between the cerebellum and the brainstem, and the spinocerebellar tracts, which relay sensory and proprioceptive information from the spinal cord to the cerebellum [14,23]. These pathways are crucial for integrating sensory feedback with motor actions. Brainstem degeneration, particularly within the pontine and olivary nuclei, can also contribute to oculomotor disturbances and postural instability. Additionally, disruptions in the cortico-ponto-cerebellar and cerebello-thalamocortical loops—connecting the cerebellum with the cerebral cortex and thalamus—contribute to both motor deficits (such as gait ataxia, dysmetria, and intention tremor) and non-motor symptoms, including cognitive impairments, emotional dysregulation, and speech difficulties [14].

PCs are particularly vulnerable to toxicity. In poly Q ataxias, toxic proteins encoded by mutated genes can easily oligomerize into aggregates that are either directly toxic or indirectly so through the sequestration of cellular components such as transcription factors and ubiquitin-specific proteases, disrupting protein homeostasis [26,27,28,29]. In contrast, non-coding repeat expansion ataxias are associated with different pathogenic mechanisms, such as the disruption of RNA homeostasis (SCA10, SCA31, SCA36, and SCA37), mutations in ion channels (SCA6, SCA13, SCA19/SCA22, SCA15/SCA16, SCA29, SCA41, SCA42, and SCA44), and repeat-associated non-AUG (RAN) translation that occurs in a different reading frame (SCA8 and SCA12). All of these mechanisms can lead to cellular toxicity [4,19,30,31,32,33].

## 3. The Essentials of MSCs and Their Characteristics

### 3.1. Basics of Stem Cell Therapy

Degenerative brain diseases are a global concern, inflicting immense suffering on numerous patients. Moreover, the non-regenerative nature of the central nervous system (CNS) poses a longstanding challenge. Although various approaches, including gene therapy and stem cell therapy, have been explored, none have yielded notably promising results so far. Nevertheless, the characteristics and potential of stem cell therapy, particularly for degenerative brain diseases, remain critical areas of focus.

Stem cell therapy can be broadly categorized into three main groups, with the first being embryonic stem cells (ESCs). ESCs are derived from the inner cell mass of the developing pre-embryo and can be propagated indefinitely in culture, having the potential to differentiate into various fully specialized cell types in the body. Such cells include neurons, cardiomyocytes, smooth muscle cells, hematopoietic cells, osteogenic cells, hepatocytes, insulin-producing cells, keratinocytes, and endothelial cells [34,35,36]. The high proliferative capacity of ESCs, along with their differentiation potential into various cell types, grants them immense clinical potential [37]. Consequently, ESCs are being utilized in numerous clinical settings and preclinical studies involving spinal injuries, cardiovascular diseases, and various neurodegenerative disorders [38,39,40].

The second group comprises induced pluripotent stem cells (iPSCs), which are primarily derived from embryonic fibroblasts and tail-tip fibroblasts in mice or dermal fibroblasts in humans. Various types of adult somatic cells can be reprogrammed to a pluripotent state, making them suitable sources for generating iPSCs [41,42]. iPSCs share similar morphology and pluripotency with ESCs, as well as the capability to form teratomas [43]. iPSCs were first derived in 2006 by Takahashi and Yamanaka through the reprogramming of adult somatic cells, such as skin fibroblasts, into a pluripotent state through the forced ectopic expression of transcription factors, including OCT4, SOX2, KLF4, c-MYC, NANOG, and LIN28 [41,44]. Due to their autologous nature, iPSCs mitigate concerns related to immunological rejection and do not require immunosuppressants. Their ESC-like differentiation capabilities make them a viable alternative to ESC-based therapeutics [43]. Additionally, iPSC technology helps overcome ethical and supply issues associated with ESCs, which can only be sourced from embryos discarded after in vitro fertilization [45]. Given these advantages, iPSCs present two major applications: cell-based transplantation therapies and the development of novel human disease models [46].

The third group is MSCs, which are capable of self-renewal and differentiation into various functional cell types, establishing them as the most clinically studied experimental cell therapy worldwide [47,48]. The primary sources of MSCs for newborns are umbilical cords and placental tissues, while in adults, they are mainly derived from the bone marrow and adipose tissues [49,50,51,52]. Notably, MSCs from different sources exhibit distinct proliferative and multilineage potentials. For instance, MSCs derived from fetal tissues demonstrate higher proliferative capacity and lower immunogenicity compared to those from adult tissues. Similarly, MSCs from the bone marrow or chorionic villi show superior pro-angiogenic characteristics than those derived from the adipose tissues and umbilical cords [53]. Furthermore, the safety of MSCs has been validated through numerous studies. A comprehensive meta-analysis by Lalu and colleagues, involving over 1000 patients, found no association between MSC administration and acute inflammation-related toxicity, organ system complications, infections, malignancies, or mortality [54]. However, to be classified as authentic human MSC-like cells, MSCs extracted from various tissues must meet the International Society for Cellular Therapy (ISCT) criteria. Specifically, MSCs should express specific surface markers, including 5′-nucleotidase (CD73), Thy-1 (CD90), and Endoglin (CD105), while lacking markers such as CD14 (macrophage marker), CD34 (hematopoietic stem cell marker), CD45 (lymphocyte marker), CD19 (B cell marker), CD79a (B-cell antigen receptor complex-associated protein alpha chain), and HLA-DR (MHC class II surface receptor). Additionally, they must demonstrate the ability to differentiate into adipogenic, osteogenic, and chondrogenic lineages in vitro [55].

This section compares different cell sources for cell therapy. ESCs are derived from early-stage embryos, rendering donor age irrelevant. However, their use poses several challenges, including ethical concerns [56,57], genomic instability, tumor formation risk, unavailability of both allogenic and autologous sources, and potential immunological rejection [58]. In contrast, iPSCs are generated from reprogrammed adult somatic cells, addressing the ethical concerns associated with ESCs and overcoming the issues of unavailable allogenic and autologous sources and immunological rejection risk. Nevertheless, iPSCs still face challenges such as genomic instability, persistent expression of reprogramming factors, and tumor formation risk, as evidenced by their ability to form teratomas in mice [43]. Additionally, the low reprogramming efficiency of iPSCs, along with their immunogenicity and heterogeneity among clones, significantly hinders their therapeutic applications [57,59]. Moreover, current technologies face limitations in producing large quantities of high-purity iPSCs due to low cell induction efficiency [58,60]. On the other hand, MSCs are relatively free from many issues associated with other stem cell therapies. However, the use of relatively large, living MSCs carries inherent risks such as microvascular occlusion, potential malignant transformation, pro-arrhythmic effects, and abnormal ossification. Additionally, operational challenges persist in monitoring and maintaining the viability and potency of cell-based MSC therapeutics during manufacturing, storage, and delivery [58,61].

Nevertheless, the ability of MSCs to differentiate into various cell types, along with the therapeutic roles of MSC-derived secretomes highlighted in recent studies, validates their potential compared to other stem cell types [61,62].

### 3.2. Characteristics of MSCs

MSCs display several important biological characteristics, such as secretome production, immunomodulation, and homing ability (Figure 1). The MSC secretome encompasses a diverse array of bioactive factors secreted by MSCs, including cytokines, chemokines, inflammatory factors, and extracellular vesicles such as exosomes and microvesicles, which may also carry miRNA [63,64]. These secreted factors mediate interactions between MSCs and surrounding tissues, promoting tissue repair and regeneration. Additionally, they provide protective functions, including anti-apoptotic, anti-inflammatory, and anti-scarring effects, along with immunomodulatory, angiogenic, and anti-tumorigenic activities [53]. While specific soluble factors in the MSC secretome slightly vary by tissue source, core cytokines such as CCL2, CCL5, bFGF, insulin-like growth factor-1 (IGF-1), IL-6, TGF-β, vascular endothelial growth factor (VEGF), and TNFR1 are present in all MSCs [63]. Leveraging these characteristics, the cell-free approach, which utilizes only the MSC secretome, is gaining attention due to its superior therapeutic outcomes compared to cell transplantation, with a minimized risk of side effects [57].

MSCs can also modulate both innate and adaptive immunity through immunosuppressive mechanisms. This is mediated by the secretion of soluble factors and direct interactions with immune cells, including the inhibition of proliferation and functions of T cells, B cells, and natural killer (NK) cells [51,65]. Recent studies have indicated that MSCs exert immunomodulatory effects through interactions with regulatory T cells (Tregs) and monocytes [66]. Sotiropoulou et al. demonstrated that MSCs can suppress NK cell proliferation through both direct interactions and the secretion of soluble molecules [67]. Moreover, when adipose-derived MSCs (AD-MSCs) were co-cultured with B cells, they inhibited B-cell proliferation and shifted the cytokine profile toward the anti-inflammatory spectrum [68]. MSCs also inhibit T-cell proliferation and co-regulate cytokine receptor expression through direct cell contact [69].

MSCs exhibit a homing ability, referring to their capacity to target and migrate to areas of damaged tissue or inflammation in the body. This characteristic is crucial for the therapeutic efficacy of stem-cell-based treatments. MSCs display varying degrees of homing affinity depending on the tissue type, mediated by the expression of different chemokine receptors [51]. Numerous studies demonstrate that transplanted MSCs tend to migrate toward inflamed or damaged tissues. However, these studies have also revealed that MSCs often migrate to a relatively small proportion of tissues in vivo, indicating low transplantation efficiency under physiological conditions. Factors such as the interval and quantity of transplantation, culture methods, and MSC pretreatment may influence this outcome [70]. Several migration factors are involved in the homing process, including SDF-1, TRAIL, RANKL, PDGF, IL-17, bFGF, INF-γ, IGF, TGF-β, EGF, and EPO. These factors are released at the site of injury by various cells (e.g., endothelial cells, tumor cells, and affected tissue cells). To respond to these factors, MSCs must express specific receptors, including CXCR4, TRAIL receptors (DR5 and DcR2), RANK, PDGF receptors (type α and β), IL-17 receptor, bFGF receptor, INF-γ receptor, IGF receptor, TGF-β receptor, EGF receptor, and EPO receptor, among others [71].

Currently, treatment for CA primarily focuses on symptom relief through physical therapy, occupational therapy, and medication [72,73,74,75,76,77]. However, these approaches have limitations in halting disease progression or promoting recovery, and few address the underlying causes. As a result, there is a growing need for novel therapeutic strategies, with stem-cell-based therapy emerging as a promising alternative. Numerous in vitro and in vivo experiments have demonstrated the vast therapeutic potential of MSCs, mediated by the secretion of extracellular vesicles, cytokines, and growth factors. These secretions modulate both adaptive and innate immune responses, promote tissue regeneration, and alleviate pathological symptoms [57]. To advance stem cell therapy, gaining deeper insights into its application across various diseases is crucial. MSCs, in particular, are being actively studied for their potential in treating neurodegenerative diseases.

## 4. Preclinical Research on MSC Treatments for CA

### 4.1. The Role of MSCs in Addressing Neurological Diseases

MSCs can be employed in treating neurological conditions through several key mechanisms. First, MSCs can migrate to damaged areas, protect tissues, and inhibit cell death. Second, they secrete neurotrophic factors that facilitate the regeneration of damaged axons. Third, MSCs contribute to neuronal protection under chronic inflammatory conditions through their immunomodulatory functions. Due to these properties, MSCs are being actively explored as a potential treatment for neurological disorders such as Alzheimer’s disease (AD), Parkinson’s disease (PD), and multiple sclerosis (MS).

MSCs have been introduced through several methods in preclinical models of AD, demonstrating neuroprotection [78,79,80]. For example, in a study using the BV-2 mouse cell line, an alternative to primary microglial cultures, MSC-derived extracellular vesicles significantly reduced the secretion of TNF-α and NO in Aβ-stimulated BV-2 cells, resulting in reduced inflammation and neuroprotective effects [81]. Similarly, in vivo studies have demonstrated the potential of MSC-derived secretomes. A study administered secretomes from MSCs exposed in vitro to AD mouse brain homogenates (MSC-CS) to 25-month-old APP/PS1 mice through intranasal (IN) delivery once weekly. This treatment was assessed using the novel object recognition test to evaluate memory recovery and measured gliosis levels and the phagocytic marker CD68 to evaluate amyloidosis. Repeated IN delivery of MSC-CS resulted in significant memory recovery, a reduction in plaques with lower densities of β-amyloid oligomers, and decreased neuroinflammation [82]. In another study, exosomes derived from MSCs or hypoxia-preconditioned MSCs were injected into APP/PS1 transgenic mouse models, leading to decreased amyloid plaques and improvements in cognitive function [83].

The beneficial effects of MSCs in vivo are primarily attributed to the increased expression of anti-inflammatory cytokines, neurotrophic factors, and antioxidant properties [84,85,86,87]. For instance, transplantation neuron-like cells derived from human umbilical cord MSCs (UC-MSCs) in AβPP/PS1 mice increased the expression of anti-inflammatory cytokines such as IL-4 while significantly reducing pro-inflammatory cytokines IL-1β and TNF-α [88]. Additionally, MSC transplantation in APP/PS1 mice reduced oxidative stress in the hippocampus, as evidenced by lower superoxide anion levels [89]. Furthermore, soluble intracellular adhesion molecule-1 (sICAM-1), derived from human umbilical cord blood-derived MSCs (hUCB-MSCs), has been shown to improve microglial function and reduce Aβ plaques by inhibiting CD40/CD40L activity in APP/PS1 mouse models [90]. In an AD rat model, MSCs exhibited anti-amyloid and anti-apoptotic effects, while microRNAs from extracellular vesicles derived from bone-marrow-derived MSCs (BM-MSCs) promoted cell survival and neuroprotection by inhibiting BACE1 and Aβ production [91,92]. Additionally, in the 5xFAD mouse model, galectin-3 secreted by hUCB-MSCs was found to regulate abnormal tau accumulation through protein–protein interactions [93]. Thus, these studies underscore the multifaceted therapeutic potential of MSCs and their secretomes in modulating inflammation, oxidative stress, and amyloid/tau pathology in AD.

The pathological features of PD include the loss of dopaminergic neurons in the substantia nigra, resulting in reduced dopamine (DA) levels in the striatum. Additionally, Lewy bodies composed of α-synuclein are observed in the surviving neurons [94,95]. MSCs and their exosomes promote neuroprotection of dopaminergic neurons and improve motor functions [58,96,97]. In murine models of PD, MSC transplantation has been shown to increase the expression of tyrosine hydroxylase (TH) and survivin and improve motor abilities [98,99]. Additionally, MSCs regulate autophagy-lysosomal activity to inhibit cell death, which is further enhanced by α-synuclein-primed MSCs that upregulate miRNAs associated with autophagy [100,101,102]. Moreover, MSC-secreted NGF, BDNF, and GDNF contribute to neuronal protection by activating the NF-κB signaling pathway, leading to the production of pigment epithelium-derived factor (PEDF), a neuroprotective molecule [103,104]. MSCs also secrete VEGF and fibroblast growth factor 20 (FGF20), which enhances neurogenesis [105,106].

However, recent studies have raised concerns about the potential spread of α-synuclein pathology in transplanted cells, as well as the challenges in restoring broader neural circuit functions [107,108]. To address these concerns, current research is focused on enhancing the therapeutic efficacy of MSCs through genetic modifications and priming techniques. These efforts aim to improve the ability of MSCs to combat α-synuclein pathology and restore dopaminergic functions [107,108].

MS is a chronic inflammatory disease of the nervous system, characterized by myelin loss due to immune system attacks [109]. T-helper 17 (Th17) cells and Tregs are critical to immune regulation. Th17 cells exhibit excessive proliferation and activation, releasing pro-inflammatory cytokines such as IL-17, which contribute to inflammation. Meanwhile, Tregs, which suppress immune responses through factors including TGF-β and CD25, are deficient in this disease [110,111]. MSCs have been extensively studied in MS models for their immunomodulatory and neuroprotective properties [112,113]. In the experimental autoimmune encephalomyelitis (EAE) mouse model, MSCs inhibit the differentiation of naive T cells into Th17 cells, promote the generation of Tregs, and reduce IL-17 and IL-22 levels, thereby enhancing immunosuppressive functions. MSCs play a significant role in reducing demyelination by inhibiting excessive T-cell activity through the upregulation of Treg cytokine expression [114,115,116]. When transplanted, bone marrow-derived MSCs (BM-MSCs) and AD-MSCs exhibited enhanced immunosuppressive capabilities and promoted remyelination by increasing the expression of FoxP3 and CD4, as well as boosting IL-10 secretion [117]. Moreover, exosomes and extracellular vesicles derived from MSCs have been demonstrated to attenuate inflammation and demyelination in murine models of MS, promoting microglial polarization toward the anti-inflammatory (M2) phenotype [118,119]. In the cuprizone (CPZ)-induced mouse model of MS, MSCs were demonstrated to reduce pro-inflammatory cytokines [120]. Here, the combination of MSC transplantation with astrocyte ablation or transcranial direct current stimulation (tDCS) improved remyelination by increasing the number of oligodendrocytes and mitigating microglial responses [109,121].

Given the characteristics of MSCs and their therapeutic potential in various neurological diseases, there is growing interest in their effects on SCA. Ongoing studies are investigating the neuroprotective effects of MSCs and their ability to delay disease progression in SCA models, providing new hope for affected patients.

### 4.2. MSCs for SCA

This section summarizes the studies that investigated the effects of MSCs or their properties in animal models of SCA (Figure 2). Two studies, those of Tsai et al. [5] and Oliveira et al. [8], demonstrated the effects of direct MSC engraftment in SCA mouse models. Moreover, three additional studies—those of Correia et al. [7], Suto et al. [6], and You et al. [9]—investigated the effects of introducing subcellular components derived from MSCs, such as secretome or exosomes, in SCA mouse models.

#### 4.2.1. Administration of MSCs Engraftment

Tsai et al. [5] explored the therapeutic effects of human MSCs derived from Wharton’s jelly in the umbilical cord by injecting them into the cerebellum of transgenic mice carrying the SCA1 gene. The injected MSCs remained localized in the cerebellum for several months without differentiating into other cell types and secreted trophic factors and cytokines, including neutrophil-activating protein-2 (NAP-2), angiopoietin-2, BDNF, CXCL-16, and platelet-derived growth factor-AA. These secretions reduced cerebellar atrophy and prevented the death of PCs. Subsequent electrophysiological studies confirmed an enhancement in neuromuscular response strength in the MSC-injected group of SCA1 mice, which also demonstrated improved motor functions across various tests.

Oliveira Miranda et al. [8] compared the effects of a single intracranial injection of MSCs against repeated systemic administration of MSCs in Tg-ATXN3-69Q Machado-Joseph disease (MJD) mice, a model of MJD/SCA3. The results showed that the single intracranial injection approach produced only transient effects, whereas repeated systemic administration led to improvements in motor behavior and the recovery of some neurometabolic defects in these mice. Additionally, periodic MSC administration was found to enhance the expression of GABA and the glutamate–glutamine complex, supporting previous studies demonstrating that MSCs can strengthen the GABAergic system. The study also confirmed a reduction in the myo-inositol level, suggesting it may serve as a biomarker for improved neuronal preservation. This research underscores the clinical relevance of continuous or repeated infusion of MSCs and its advantages over the single injection approach. The study also emphasizes the need for validation of the long-term safety and efficacy of MSC therapy.

#### 4.2.2. Administration of MSCs-Derived Subcellular Components

Correia et al. [7] used the CMVMJD135 mouse model of SCA3/MJD to evaluate the therapeutic potentials of human MSC transplantation and MSC secretome injections. These treatments were administered to the cerebellum, striatum/substantia nigra, and spinal cord, with various behavioral tests conducted to assess phenotypic outcomes. The study aimed to compare the therapeutic efficacy of both approaches and determine which injection site offers the greatest benefit for motor and behavioral improvements. When human mesenchymal stem cells (HMSCs) were injected into the cerebellum, no significant effects were observed; however, a slight improvement was noted following injection into the spinal cord. In contrast, while hMSC secretome injections into the striatum/substantia nigra or spinal cord yielded no noticeable effects, a more pronounced therapeutic benefit was observed with injections into the cerebellum. The MSC secretome-treated model exhibited greater improvements compared to the MSC-treated model despite initially presenting more severe behavioral deficits. Additionally, the study suggested that repeated MSC injections are likely to yield more sustained effects over time compared to a single injection, and a multi-regional treatment approach may be more beneficial than targeting a single area. In conclusion, the respective study demonstrated the preclinical therapeutic efficacy of both MSCs and MSC secretome, guiding future research directions.

Furthermore, Suto et al. [6] conducted a follow-up study to determine whether the effects observed in their previous experiment [122]—where intrathecal injection of MSCs alleviated peripheral nervous system (PNS) neurodegeneration in SCA1-knock-in mice carrying a mutant ATXN1 gene—could be reproduced using MSC-conditioned medium (MSC-CM). The group receiving MSC-CM demonstrated improvements in morphological pathologies, such as axonal and myelin degeneration. Additionally, reduced nerve conduction velocities in spinal motor neurons and motor incoordination were also ameliorated. This confirms the therapeutic effects of the MSC secretome derived from the paracrine activity of MSCs, suggesting the potential for a cell-free therapeutic method for treating SCA1 with minimized risks of unexpected side effects associated with direct MSC introduction. The authors hypothesized that hepatocyte growth factor (HGF), a component of the MSC secretome, may play a role in maintaining the expression levels of glutamate/aspartate transporter (GLAST) and glutamate transporter-1 (GLT-1) in the cerebellar cortex, thereby reducing excitotoxicity and the subsequent degeneration of PCs in the cerebellum.

You et al. [9] investigated the potential therapeutic effects of MSC-derived exosomes in YACMJD84.2 mice, a model of SCA3. MSC-derived exosomes present several advantages over direct MSC injection in various therapeutic studies targeting brain diseases, including low immunogenicity, extended circulation half-life, the ability to cross the blood–brain barrier (BBB), and potential roles in mediating regenerative responses [61,62]. The exosomes were administered through tail vein injection, while the study assessed improvements in motor dysfunction and neuropathological changes. Based on prior findings that MSC treatment after disease onset failed to alleviate the associated pathology, and a pre-treatment approach was adopted to explore the potential efficacy of exosomes [123]. The treatment increased the expression of the anti-apoptotic Bcl-2 protein relative to the pro-apoptotic Bax protein, resulting in a reduction in PC loss in the cerebellum. Additionally, the treatment reduced gliosis, a condition where neuroglial cells enlarge in response to neuronal damage caused by the exosomes’ anti-inflammatory properties. However, the levels of mutant ATXN3 remained unchanged compared to the control group, indicating that MSC exosomes do not directly inhibit the mutant ATXN3 protein.

## 5. Clinical Trials for MSC-Based SCA Therapy

### 5.1. Review of Previous Clinical Trials

Over the last few decades, advancements in genetic testing technologies have significantly increased the diagnostic potential for neurogenetic disorders [19,124]. Additionally, improvements in stem cell extraction and stable delivery have brought greater attention to stem-cell-based therapies. In this context, reviewing past clinical trials is essential to understand the strategies of current studies. Various clinical trials have explored the application of MSCs as a treatment strategy for neurological diseases such as AD, PD, and MS [125,126]. For instance, in AD patients, injections of 3.0 × 10^6^ and 6.0 × 10^6^ umbilical cord MSCs (UC-MSCs) into the hippocampus and thalamus resulted in an improvement in neurological symptoms, with no serious adverse events observed [127]. In MS patients, intrathecal injection of autologous BM-MSCs led to enhanced neurological functions, suppression of lymphocyte proliferation, and improved visual symptoms [128]. Additionally, a randomized, double-blind, single-center, phase 2 study is currently in progress to assess the efficacy and safety of AD-MSCs for the treatment of PD (ClinicalTrials.gov: NCT04995081). Overall, MSC administration is safe in treating various neurological diseases. However, studies testing the safety and efficacy of MSCs in SCA patients are still limited. A recent systematic review and meta-analysis noted that clinical research utilizing stem cells for SCA requires larger sample sizes and placebo-controlled studies [129].

The most recent clinical trial for SCA was a pilot, open-label, phase I/IIa study reported in 2017 and conducted in Taiwan [130]. In this study, six SCA3 patients aged 20 to 70 years received intravenous infusions of 7 × 10^7^ AD-MSCs and were monitored for one year, during which no MSC-related adverse effects were observed. Furthermore, evaluations based on the Scale for the Assessment and Rating of Ataxia (SARA) scores, sensory organization testing (SOT), oculomotor testing, magnetic resonance spectroscopy (MRS), and 18F-fluorodeoxyglucose (18F-FDG) positron emission tomography (PET) indicated initial, albeit transient, improvements in the test results along with increased glucose metabolism in some patients. The SARA score used in this evaluation is a reliable and validated tool for assessing ataxia, serving as an effective primary outcome measure in clinical trials. It comprises eight components: (1) gait (scored 0 to 8), (2) stance (scored 0 to 6), (3) sitting (scored 0 to 4), (4) speech disturbance (scored 0 to 6), (5) finger chase (scored 0 to 4), (6) nose–finger test (scored 0 to 4), (7) fast alternating hand movements (scored 0 to 4), and (8) heel–shin slide (scored 0 to 4). The total score ranges from 0 (no ataxia) to 40 (most severe ataxia) [131]. Although the study was limited by the absence of a placebo-controlled group due to the small patient cohort, the safety and tolerability of 7 × 10^7^ AD-MSCs were confirmed [130]. Previous studies involving 16 and 14 SCA patients who received UC-MSCs via intravenous and intrathecal injections, respectively, reported similar positive outcomes. However, these studies were also constrained by small sample sizes and non-randomized, open-label designs [132,133].

The varying effects of stem cell sources, patient age, and transplantation methods have yet to be sufficiently validated for stem-cell-based SCA therapy. Moreover, the lack of placebo-controlled, double-blind trials indicates the need for more rigorous clinical studies. Future clinical trials should aim to address these limitations and focus on assessing the safety and tolerability of MSC-based SCA therapy, which remains elusive given the limited number of trials conducted for this condition.

### 5.2. Current and Upcoming Clinical Trials

Although stem-cell-based SCA therapy has shown promising results in reducing cerebellar degeneration in preclinical animal models, there is insufficient clinical evidence to validate its efficacy in humans [7,129,134]. The safety and efficacy of MSC-based treatments must be verified in clinical trials involving humans to optimize the source of MSCs, transplantation method, and appropriate number of cells for injection.

Currently, two phase 2 clinical trials are scheduled to begin, although they are listed as “not yet recruiting” on ClinicalTrials.gov (ClinicalTrials.gov: NCT06397274; NCT03378414) (Table 1). These trials will use allogeneic AD-MSCs and human UC-MSCs, administered via intravenous infusion and intrathecal injection, respectively. The study utilizing AD-MSCs will be a randomized, double-blind, placebo-controlled, single-center trial involving 20 participants diagnosed with SCA3. These participants, aged between 20 and 70 years, will have SARA scores ranging from 5 to 15. The SARA score increases with the progression of SCA, with scores between 5 and 15 corresponding to stages 1 and 2 [131]. Disease progression is classified from stage 0 to stage 4: stage 0 indicates no gait difficulties; stage 1 marks the onset of gait difficulties; stage 2 represents the loss of independent gait; stage 3 involves confinement to a wheelchair; and stage 4 results in death [135]. The participants will be divided into two groups: the MSC intravenous infusion group and the placebo intravenous infusion group. The primary outcome will be assessed by the changes in SARA scores 6 months after the treatment (ClinicalTrials.gov: NCT06397274). Similarly, the study utilizing UC-MSCs will be a randomized, open-label, parallel-controlled trial involving 45 participants aged between 16 and 60 years. This cohort will include patients clinically and genomically diagnosed with SCAs, with SARA scores between 2 and 5, and who are capable of completing an 8 m walking test. This trial will include three groups: the intravenous infusion group, the intrathecal injection group, and the control group. Each participant will receive 2 × 10^7^ cells, and the primary outcome will be assessed by the changes in SARA scores 12 months after the treatment. Secondary outcomes will be assessed by the changes in MRI results, Inventory of Non-Ataxia Symptoms (INAS) scores, and cerebrospinal fluid (CSF) analyses (ClinicalTrials.gov: NCT03378414).

These upcoming trials aim to address the limitations of previous studies, refining key variables such as cell delivery, differentiation efficiency, and immune responses to MSCs. Successfully resolving these issues will assist in developing safer and more effective MSC treatments for SCAs and potentially other neurological conditions. Moving forward, these clinical trials represent an essential step in overcoming the unmet therapeutic needs of SCA patients, providing hope for more effective interventions.

### 5.3. Future Directions for MSC-Based Treatments

The treatment utilizing MSCs leverages their ability to migrate to damaged areas and their reported capability to traverse the blood–brain barrier [126,136]. Upon migration, MSCs secrete cytokines and neurotrophic factors that modulate inflammation and promote neuroprotection [137]. Although MSCs can differentiate into other mesenchymal lineages to repair neural damage, clinical studies indicate that the number of MSCs differentiating into neural cells is relatively low. Consequently, symptom improvement is primarily attributed to the release of neurotrophic factors rather than direct cellular integration [138]. However, complications such as pulmonary vein embolism have been observed in some cases, particularly with high-dose intravenous infusion of AD-MSCs [139,140]. Additionally, given the complex pathology of SCA and the fact that many mouse models do not fully recapitulate the human condition, delivering MSCs precisely to the target site in human patients could prove more challenging than in murine models. This highlights the need for further investigation and development of cell delivery methods to enhance accuracy and efficacy.

One of the key challenges in MSC-based therapy is the variability in clinical trial protocols, such as differences in cell sources, dosages, and intervals. To enhance the therapeutic efficacy of MSCs and address these challenges, several strategies have been proposed, including priming MSCs and employing combined approaches, such as MSC-derived neural progenitors or co-grafting with fetal ventral mesencephalic cells [102,141]. These methods aim to improve MSC migration to damaged areas while promoting differentiation into local neural cells, improving functional integration and recovery [142,143,144]. The use of biomaterials, such as immuno-suppressive hydrogels, to enhance MSC survival post-transplantation presents another promising avenue. Hydrogels with immunosuppressive properties can support MSC survival, reduce inflammation, and improve regeneration. Further exploration of these environments could maximize the therapeutic potential of MSC-based treatments [145].

Another challenge is managing potential immune rejection of the stem cells derived from certain sources, which may limit their therapeutic efficacy. Also, preventing pulmonary embolism, which can result from intravenous administration of MSCs, needs to be carefully considered. This may involve strategies including heparin treatment or optimizing dosages and infusion protocols [7,142]. While intrathecal injection of MSCs could circumvent some issues associated with intravenous administration, further research is required to ensure this method does not introduce new complications.

The use of secretomes, including exosomes and extracellular vesicles, in MSC therapy is critical due to their lower potential for immune rejection. However, a significant obstacle to exosome-based therapies is the challenge of isolating sufficient amounts of exosomes. Although ultracentrifugation is the most common method for exosome isolation, it is time-consuming and has drawbacks, such as potential sample contamination and damage to the structure and components of exosomes [146]. Compared to ultracentrifugation, anion exchange chromatography yields nearly twice the number of exosomes while requiring 20% of the time needed for ultracentrifugation [147]. You et al. [9] consistently obtained high-quality exosomes from MSCs using anion exchange chromatography, demonstrating substantial potential to meet the demands of clinical applications.

Clinically, the variable outcomes of stem cell source, patient age, and transplantation method have not been adequately validated in stem-cell-based treatments for SCA [129]. Therefore, future clinical trials aim to refine key variables such as cell delivery, differentiation efficiency, and immune responses to MSCs [ClinicalTrials.gov: NCT06397274, NCT03378414]. Nevertheless, given the limited number of clinical trials conducted for this condition, prioritizing the evaluation of the safety and tolerability of these treatments is crucial.

Given the limitations of current animal models, which may not fully replicate human neural structures and metabolic processes, developing more advanced models that closely mimic human neurobiology is critical. This improvement will enable more accurate pre-clinical assessments of MSC therapy before transitioning to human clinical trials. While MSC-based treatments still face numerous challenges, overcoming them could lead to breakthroughs for not only SCA but other neurodegenerative diseases.

## 6. Conclusions

This review summarizes the characteristics and mechanisms of MSCs and their applications in animal models of neurodegenerative diseases. It discusses the current status of clinical trials involving MSC-based SCA therapy and suggests future directions based on previous findings. Stem-cell-based therapeutics are highlighted for their capability to migrate, modulate the immune system, differentiate, and regenerate axons, all of which contribute to promoting neuroprotective effects. In various animal models of neurological diseases, including SCA, MSC-based therapeutics have yielded positive results. Nevertheless, due to the complexity of the pathological mechanisms underlying SCA, developing more advanced animal models that closely replicate human diseases is essential. Additionally, the pitfalls arising from the gap between treatments administered before and after lesion onset highlight the need for further animal studies that initiate treatment after the disease progression.

Completed clinical trials of MSC-based SCA therapy have been limited by various factors. Future trials must utilize larger sample sizes and minimize bias to thoroughly validate the safety, tolerability, and optimal dosage of MSCs. Furthermore, exploring strategies to enhance MSC differentiation and improve their delivery will be imperative for achieving successful treatment outcomes. Despite these challenges, MSC therapy holds significant potential for treating various neurological disorders and providing relief to affected patients.

## Figures and Tables

**Figure 1 biomedicines-12-02507-f001:**
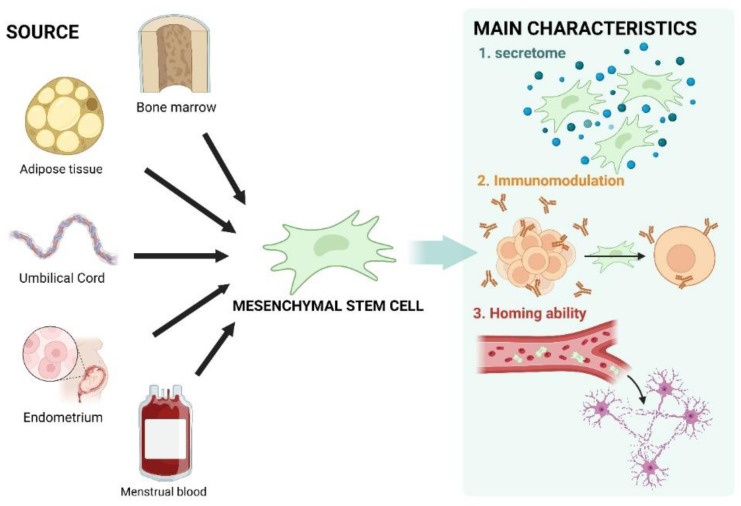
Schematic illustration showing MSC sources and key characteristics. MSCs can be isolated from multiple tissues, including bone marrow, adipose tissue, umbilical cord, endometrium, and menstrual blood. They have unique properties, such as promoting paracrine activity via secretome secretion and modulating immune responses to protect themselves and other cells. Additionally, their homing ability allows them to target damaged areas. This figure was created using BioRender.com (agreement number: WW27CJ7DXJ).

**Figure 2 biomedicines-12-02507-f002:**
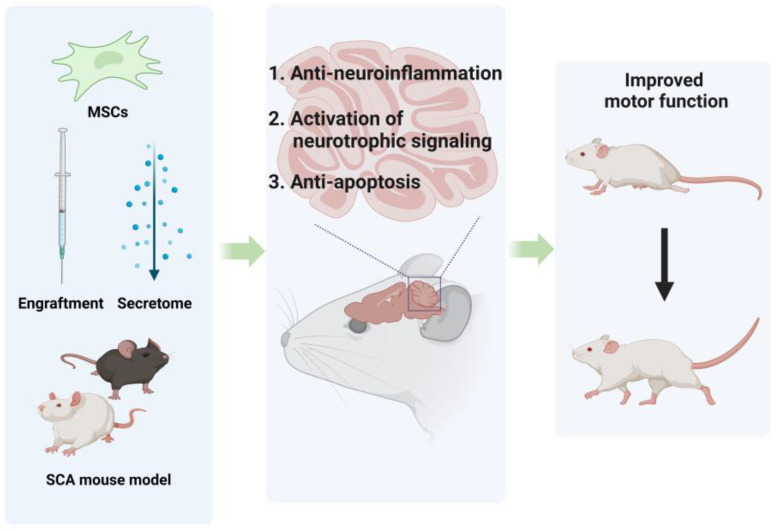
Schematic illustration of the experimental potential of MSCs for SCA therapy in vivo. Current efforts focus on leveraging MSCs and their properties to treat SCA, as depicted in the illustration. In SCA mouse models, MSCs or MSC-derived secretomes have shown anti-inflammatory effects and activation of neurotrophic signaling through the production of anti-inflammatory cytokines and neurotrophic factors. Additionally, anti-apoptotic effects were observed following the administration of MSC-derived exosomes. Although further research is needed for clinically effective treatment, these recent findings clearly demonstrate the therapeutic potential of MSCs for SCA. This figure was created using BioRender.com (Agreement number: IZ27G8YU7I).

**Table 1 biomedicines-12-02507-t001:** Summary of upcoming studies.

ClinicalTrials.gov ID	NCT03378414	NCT06397274
Study Start	31 December 2024	1 June 2025
Phase	Phase 2	Phase 2
Details	Randomized, open label, and parallel controlled experiment; Follow-up visit by doctors 1, 2, 3, 6, and 12 months after treatment, and efficacy evaluation employed.	Randomized, double-blind, placebo-controlled, single-center study.
Enrollment	45	20
Ages Eligible for Study	16 years to 60 years (child, adult)	20 years to 70 years (adult, older adult)
Inclusion Criteria	Spinocerebellar ataxias (SCA); SARA ^1^ scores of 2–5; Can complete 8 m walking test; No stem cell treatment in 6 months; Signed the consent form based on the experiment process and statement.	Genotypically confirmed SCA3; SARA ^1^ scores of 5–15; Female subjects of child-bearing potential and are capable of conception must be post-menopausal; Male subjects must use a medically accepted form of contraception during the study period; Signed informed consent.
MSC source	Human umbilical cord mesenchymal stem cells.	Stemchymal^®^ (allogeneic adipose-derived mesenchymal stem cells).
Participant Group	Intravenous infusion group: 2 × 10^7^ cells (30 mL); Intrathecal injection group: 2 × 10^7^ cells (1 mL); Control groups.	Experimental: Stemchymal^®^ through intravenous infusion; Placebo comparator: placebo through intravenous infusion.
Outcome Measures	Primary: SARA ^1^ score; Secondary: MRI plain scan of brain, INAS ^2^ score, cerebrospinal fluid routine.	SARA ^1^ scores from baseline (week 0) to 6 months (week 24).

^1^ SARA score: Scale for the Assessment and Rating of Ataxia, ^2^ INAS score: Inventory of Non-Ataxia Symptoms.

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
