# Peer review of "The Potential of Mesenchymal Stem Cells in Treating Spinocerebellar Ataxia: Advances and Future Directions"

_biomedicines, 2024, doi:10.3390/biomedicines12112507_

Round 1

Reviewer 1 Report

Comments and Suggestions for Authors

This review discusses the advances made in using mesenchymal stem cells as a potential therapeutic approach in the treatment of spinocerebellar ataxias with a focus on research in animal animals and early-stage clinical trials in patients.  In addition, there is information of the use of these cells in other neurodegenerative diseases and their animal models.  The material is presented clearly and pitfalls in the approach are discussed.

There are a few additional points that should be addressed.  One is that many of the hereditary ataxias, particularly the polyglutamine-related SCAs, have quite complex pathologies that involve not only the cerebellum cortex, but the deep cerebellar nuclei, the basal pons, inferior olivary  nuclei, spinocerebellar tracts, and often both motor and sensory cranial nerve nuclei as well as spinal motor neurons.  Some have striatal and probably cerebral cortical involvement as well.   Therefore, getting the cells to the proper locations in a human patient may be a significantly greater problem than it is in murine models, many of which are not truly replications of the human disease.  Secondly, although the authors refer to the effects of stem cells as being “regenerative”,  the data showing benefit in the murine models is derived from mutant animals treated in the pre-symptomatic phase of the disease.  This finding suggests that it may slow the progression of the disease, but not truly “regenerate” what is lost.  Conditional transgenic models have shown that there is a window in which turning off the mutant transgenic works only up to a certain point in the progression of the disease and becomes less effective as the pathological changes accrue.  The implications of this for human treatment are important, since most patients are not diagnosed in the pre-symptomatic stages of the disease.  This point should be raised in the discussion of possible pitfalls and calls for further animal studies where treatment is initiated later in the course of the disease.

A few other minor points.

Line 42:  There is increased survival but no evidence for proliferation of PCs in this study, as there is no cellular loss prior to the initiation of treatment.   

Line 56:  In most ataxias there is more than pathology in Purkinje cells (see above comment) and a number of the symptoms are not directly related to degeneration in the cerebellum or its primary connections.

Lines 77-78:  This sentence is an oversimplification of the complexity of the pathology.

Line 350: Give a citation for the previous studies that show no effect in symptomatic mice.

Line 473:  See the comment above on the use of “neuroregeneration”.

Author Response

Response to Reviewer 1:

This review discusses the advances made in using mesenchymal stem cells as a potential therapeutic approach in the treatment of spinocerebellar ataxias with a focus on research in animal animals and early-stage clinical trials in patients.  In addition, there is information of the use of these cells in other neurodegenerative diseases and their animal models.  The material is presented clearly and pitfalls in the approach are discussed.

Response: We thank the reviewer for the comprehensive evaluation of our manuscript and for providing constructive comments and suggestions for its revision.

Comment 1: There are a few additional points that should be addressed.  One is that many of the hereditary ataxias, particularly the polyglutamine-related SCAs, have quite complex pathologies that involve not only the cerebellum cortex, but the deep cerebellar nuclei, the basal pons, inferior olivary nuclei, spinocerebellar tracts, and often both motor and sensory cranial nerve nuclei as well as spinal motor neurons.  Some have striatal and probably cerebral cortical involvement as well.   Therefore, getting the cells to the proper locations in a human patient may be a significantly greater problem than it is in murine models, many of which are not truly replications of the human disease.  

Response: We appreciate your valuable comment. We have provided a more detailed explanation of the complex pathological mechanisms of SCA in Section 2. Additionally, we have highlighted the importance of developing animal models that closely resemble human patients, as the current models do not fully replicate these characteristics in the revised manuscript (Lines: 521-525; 563-568).

Comment 2: Secondly, although the authors refer to the effects of stem cells as being “regenerative”, the data showing benefit in the murine models is derived from mutant animals treated in the pre-symptomatic phase of the disease.  This finding suggests that it may slow the progression of the disease, but not truly “regenerate” what is lost.  Conditional transgenic models have shown that there is a window in which turning off the mutant transgenic works only up to a certain point in the progression of the disease and becomes less effective as the pathological changes accrue.  The implications of this for human treatment are important, since most patients are not diagnosed in the pre-symptomatic stages of the disease.  This point should be raised in the discussion of possible pitfalls and calls for further animal studies where treatment is initiated later in the course of the disease.

Response: Thank you again for your insightful comment. As suggested, we noted the discrepancy between preclinical studies that initiate treatment before lesion onset and the reality that most patients are diagnosed only after the disease has progressed. In the conclusion, we emphasized the need for additional animal studies that begin treatment after disease progression to address this issue (Lines 565-568). Additionally, we have modified the term 'regeneration' to more appropriate expressions (Lines 287; 326-328; 501-503; 562).

A few other minor points.

Comment 1: Line 42: There is increased survival but no evidence for proliferation of PCs in this study, as there is no cellular loss prior to the initiation of treatment.   

Response: Thank you once again for your thorough review and feedback. We have revised the manuscript based on a careful review of the references (Lines: 43-45).

Comment 2: Line 56: In most ataxias there is more than pathology in Purkinje cells (see above comment) and a number of the symptoms are not directly related to degeneration in the cerebellum or its primary connections.

Response: Following Comment 3, we appreciate your important comment. To clarify and specify the content further, we have supplemented the relevant information in the revised manuscript (Lines: 58-64).

Comment 3: Lines 77-78: This sentence is an oversimplification of the complexity of the pathology.

Response: To further clarify and specify the content, we have provided additional explanations regarding the complexity of the pathology in the revised manuscript (Lines: 81-95).

Comment 4: Line 350: Give a citation for the previous studies that show no effect in symptomatic mice.

Response: As you suggested, we have added a citation for previous studies that demonstrate no effect in symptomatic mice (Line: 408).

Comment 5: Line 473:  See the comment above on the use of “neuroregeneration”.

Response: Thank you again for your comment. We have changed 'neuroregeneration' to 'neuroprotective effects’ (Lines: 562).

The revised sections in the main text have been highlighted in red. I would like to thank the reviewers once again for their comprehensive review of our manuscript and for their important comments.

Reviewer 2 Report

Comments and Suggestions for Authors

This manuscript reviews the therapeutic potential of mesenchymal stem cells (MSCs) in treating spinocerebellar ataxia (SCA), a neurodegenerative disorder with no FDA-approved treatments. The manuscript explores how MSCs may promote neuronal growth, synaptic connectivity, and reduce inflammation. It also provides an overview of MSCs' mechanisms, including their immunomodulatory and neuroprotective effects, focusing on preclinical and clinical studies involving MSCs.

The manuscript is well designed and gives useful information.

·       The abstract is very non informative it must include the recent updates it the field and future perspectives

·       Enrich the manuscript by elaboration in the role of extracellular vesicles from MSCs in treatment of SCA

·       Modify Table 1 legend to be more informative and descriptive for the content

·       Section 4.2. must be redivided into several parts. There must be clear description and separation between the application of while cells or parts of it e.g. exocomes or other subcellular components

·       While the manuscript outlines the promise of MSC therapy for SCA, the section on future research directions could be more specific.

·       Avoid using the words “Numerous studies have demonstrated the……………..”. Instead elaborate into these srudies in scientific approach

Comments on the Quality of English Language

Minor spell checks and grammar required

Author Response

Response to Reviewer 2:

This manuscript reviews the therapeutic potential of mesenchymal stem cells (MSCs) in treating spinocerebellar ataxia (SCA), a neurodegenerative disorder with no FDA-approved treatments. The manuscript explores how MSCs may promote neuronal growth, synaptic connectivity, and reduce inflammation. It also provides an overview of MSCs' mechanisms, including their immunomodulatory and neuroprotective effects, focusing on preclinical and clinical studies involving MSCs. The manuscript is well designed and gives useful information.

Response: We sincerely thank the reviewers for their thorough evaluation of our manuscript and for providing constructive comments and suggestions. As recommended, we have supplemented and revised several sections accordingly.

Comment 1:   The abstract is very non informative it must include the recent updates it the field and future perspectives

Response: Based on the suggestion, we added a sentence at the end of the abstract section to include the current clinical status (Lines: 19-20).

Comment 2:   Enrich the manuscript by elaboration in the role of extracellular vesicles from MSCs in treatment of SCA

Response: Thank you for your valuable comment. We have provided more information about the role of extracellular vesicles from MSCs in SCA in the 'Administration of MSC-Derived Subcellular Components' section (Lines: 401-404).

Comment 3: Modify Table 1 legend to be more informative and descriptive for the content.

Response: As suggested, we updated and modified Table 1.

Comment 4:   Section 4.2. must be redivided into several parts. There must be clear description and separation between the application of while cells or parts of it e.g. exocomes or other subcellular components.

Response: Thank you for your kind suggestion. We have divided Section 4.2 into several parts and revised the paragraph throughout the entire section in the updated manuscript (Lines: 331-335; 345; 368).

Comment 5:   While the manuscript outlines the promise of MSC therapy for SCA, the section on future research directions could be more specific.

Response: Thank you once again for the comprehensive review of our manuscript. As suggested, we have added more details about future research directions in the 'Future Directions for MSC-Based Treatments' section 5.3 (Lines: 508~513; 533~549).

Comment 6:   Avoid using the words “Numerous studies have demonstrated the……………..”. Instead elaborate into these studies in scientific approach

Response: Thank you for your kind suggestion. We changed and modified this sentence (Lines: 251-283).

Comment 7:   Minor spell checks and grammar required.

Response: We conducted a spell and grammar check on the entire revised manuscript through an English correction service.

The revised sections in the main text have been highlighted in red. I would like to thank the reviewers once again for their comprehensive review of our manuscript and for their important comments.

Reviewer 3 Report

Comments and Suggestions for Authors

I appreciate the opportunity to review this article. The topic is interesting and relevant, in addition the article is well written and achieves the proposed objectives. Below are a few comments that can further improve the article quality.

Introduction

Line 39 “The therapeutic application of MSCs in SCA animal models is being extensively explored in preclinical studies.” Add references

3.1 Ataxia can be inherited, acquired, or sporadic

Line 107 Describe iPSCs sources

Add a paragraph comparing stem cell therapy types.

4.1. The Role of MSCs in Addressing Neurological Diseases

Is the information cited based on animal or cellular models? Make these informations clear throughout the session.

Line 383 Give more information about SARA score

Author Response

Response to Reviewer 3:

I appreciate the opportunity to review this article. The topic is interesting and relevant, in addition the article is well written and achieves the proposed objectives. Below are a few comments that can further improve the article quality.

Response: We thank the reviewer for bringing this to our attention. The error has been corrected accordingly.

Comment 1:  Introduction: Line 39 “The therapeutic application of MSCs in SCA animal models is being extensively explored in preclinical studies.” Add references

Response: As suggested, we added more references in this sentence (Lines: 43)

Comment 2:   3.1 Basics of Stem Cell Therapy: Line 107 Describe iPSCs sources.

Response: As suggested, we described iPSCs sources (Lines: 124-127).

Comment 3:   3.1 Basics of Stem Cell Therapy: Add a paragraph comparing stem cell therapy types.

Response: Thank you for your important comment. As suggested, we have added a paragraph comparing different types of stem cell therapies (Lines: 160-181).

Comment 4:   4.1. The Role of MSCs in Addressing Neurological Diseases: Is the information cited based on animal or cellular models? Make these informations clear throughout the session.

Response: Thank you again for your valuable comment. In this paper, we present information primarily based on animal models, and we also discuss the positive role of MSCs in cellular models. Detailed information is provided in the revised manuscript (Lines: 255-283; 288; 310; 318; 320).

Comment 5:  Line 383 Give more information about SARA score.

Response: As suggested, we have added the information about SARA score in the revised manuscript (Lines: 442-448; 472-477).

The revised sections in the main text have been highlighted in red. I would like to thank the reviewers once again for their comprehensive review of our manuscript and for their important comments.

Round 2

Reviewer 1 Report

Comments and Suggestions for Authors

The authors have adequately addressed my criticisms from the initial review.